# Pesticide Importation in Sierra Leone, 2010–2021: Implications for Food Production and Antimicrobial Resistance

**DOI:** 10.3390/ijerph19084792

**Published:** 2022-04-15

**Authors:** Raymonda A. B. Johnson, Katrina Hann, Amara Leno, Collins Timire, Alpha J. A. Bangura, Margaret I. Z. George, Hayk Davtyan, Srinath Satyanarayana, Divya Nair, Alie H. D. Mansaray, Fatmata I. Bangura, Joseph S. Kanu, Jeffrey K. Edwards

**Affiliations:** 1Crop Protection Unit, Ministry of Agriculture and Forestry, Brookfields, Freetown 00232, Sierra Leone; 2Crop Protection Department, School of Agriculture, Njala University, Bo 00232, Sierra Leone; 3Sustainable Health Systems, Freetown 00232, Sierra Leone; hann.katrina@gmail.com; 4Emergency Center for Transboundary Animal Diseases, Food and Agriculture Organisation of United Nation, Freetown 00232, Sierra Leone; lenoamara87@yahoo.fr; 5International Union against TB and Lung Disease (The Union), 75006 Paris, France; collins.timire@theunion.org (C.T.); ssrinath@theunion.org (S.S.); divya.nair@theunion.org (D.N.); 6Unique Solutons Company Limited, Freetown 00232, Sierra Leone; albangs04@yahoo.com; 7Magjay Co., Ltd., Freetown 00232, Sierra Leone; magaretigeorge@yahoo.com; 8Tuberculosis Research and Prevention Center, Yerevan 0014, Armenia; haykdav@gmail.com; 9Crops Division, Ministry of Agriculture and Forestry, Ground Floor, West Wing, Youyi Building, Brookfields, Freetown 00232, Sierra Leone; aliemans270@gmail.com; 10Epidemiology Unit—Livestock and Veterinary Services Division, Ministry of Agriculture and Forestry, Brookfields, Freetown 00232, Sierra Leone; turaybangfatmataisatu@gmail.com; 11Department of Community Health, Faculty of Clinical Sciences, College of Medicine and Allied Health Sciences, University of Sierra Leone, Freetown 00232, Sierra Leone; samjokanu@yahoo.com; 12Department of Global Health, University of Washington, Seattle, WA 98105, USA; jeffreykedwards@gmail.com

**Keywords:** agriculture, food security, operational research, plant health, plant product, SORT-IT

## Abstract

There are no previous studies reporting the type and quantity of pesticides for farming from Sierra Leone and the impact of Ebola or COVID-19 on importation. This study reviewed imported farming pesticides by the Sierra Leone, Ministry of Agriculture and Forestry (MAF), between 2010–2021. It was a descriptive study using routinely collected importation data. We found the MAF imported pesticides for farming only during 2010, 2014 and 2021, in response to growing food insecurity and associated with Ebola and COVID-19 outbreaks. Results showed insecticide importation increased from 6230 L in 2010 to 51,150 L in 2021, and importation of antimicrobial pesticides (including fungicides) increased from 150 kg in 2010 to 23,560 kg in 2021. The hazard class risk classification of imported pesticides decreased over time. Increasing amounts of imported fungicides could increase the risk of future fungal resistance among humans. We found that in responding to escalating food insecurity, the government dramatically increased the amount of pesticide importation to improve crop production. Further support is necessary to decrease the risk of worsening food shortages and the possible threat of emerging antimicrobial resistance. We recommend continued monitoring and surveillance, with further studies on the most appropriate response to these multiple challenges.

## 1. Introduction

With the increasing growth in the world population, there is a corresponding increased demand for food production. Global climate changes and pandemics, such as COVID-19, have negatively impacted food security, including crop production [1]. Because 40–50% of yearly worldwide crop production is lost secondary to pest infestations, pesticide use is critical in sustaining adequate food supplies [2].

Pesticides that contain antimicrobial substances (insecticides, fungicides) can lead to an increase in antimicrobial resistance (AMR) in the environment directly, and in humans and animals indirectly. AMR has recently been shown to account for more global deaths than HIV and malaria collectively, with the highest age-adjusted mortality in Western sub-Saharan African countries, such as Sierra Leone [3]. Despite the significant benefits of pesticides, excessive and improper use can lead to numerous adverse consequences on food security, health risks to humans and animals, and major negative economic and social effects [1,4]. There is evidence linking some pesticides to long-term severe effects on health and the environment [1]. Improper use and storage of pesticides can cause injury to both handlers and unknowingly exposed children. There has also been an increase in pesticides being used for suicides in developing countries [1].

To reduce pesticide risks, countries must wisely manage their use of pesticides. Countries must phase out the use of highly hazardous pesticides and replace them with less hazardous options, but a clear understanding of the hazard classes of pesticides is required [5]. Low- and middle-income countries (LMICs) require capacity strengthening for pesticide risk regulation and to be successful, need to ensure the availability of quality regulatory risk data [1].

It is crucial for countries to build the competency to routinely monitor pesticide use, particularly for pesticides that have antimicrobial properties. The World Health Organization (WHO) Global Action Plan on AMR recognises that routine monitoring is required on the levels, trends in sales, and application of pesticides [6]. In addition, the Food and Agriculture Organisation (FAO) Action Plan on AMR has recommended that countries need to develop the capacity to implement and monitor the adoption of international guidelines for antimicrobial use [7].

As of 2019, 93 pesticides have been approved for use in the West African region, of which 8% (7/93) have antimicrobial properties. None are extremely hazardous, 16% (15/93) are highly hazardous, and 60% (56/93) are moderately hazardous pesticides [8]. However, Sierra Leone has yet to initiate this approval process [8]. Pesticide usage in sub-Saharan African countries is thought to be limited, with a reported average of 16% of farming households utilising these agrochemicals. However, their use is increasing [9].

There is limited research on pesticides in Africa [10], and the few studies conducted in Sierra Leone do not quantify usage [11]. In addition, monitoring systems for pesticide imports, including distribution and use, remain limited. Evidence is lacking on the quantification of pesticides, including those that are hazardous and those with antimicrobial properties, in the Sierra Leonean context. 

The 2014–2016 Ebola outbreak in West Africa had a significant impact on government and economic activities in Sierra Leone [12]. Before Ebola, the government directly imported, sold and distributed agricultural inputs (seeds, fertilizers and pesticides). During the outbreak, the government reallocated resources, including those slated for the procurement of pesticides, towards the fight against Ebola virus disease. Currently, the government is employing a new strategy of gradually transitioning to a private sector-led inputs market. However, the government decided to procure further inputs to improve access by farmers in 2021, as the transition process to private sector management remains in development.

Understanding the change in quantities of pesticides for use in crop production, their hazard class and their antimicrobial class would allow for a clearer picture of the impact of the Ebola outbreak on antimicrobial use and the potential future risk of increasing AMR in Sierra Leone. We hypothesised that the importation of pesticides would be reduced during the Ebola period compared to both the before- and after- periods. In this study, we aimed to describe and quantify pesticides imported into Sierra Leone by the Ministry of Agriculture & Forestry (MAF) before (2010), during (2014) and after the Ebola epidemic (2021) by their antimicrobial and hazard classes. These three years were the only times in which the MAF imported pesticides for farming use between 2010 and 2021. 

## 2. Materials and Methods

### 2.1. Study Design and Setting

This was a descriptive study using routinely collected data. Crop farming has increased in Sierra Leone from 625,679 hectares in 2004 to 1,021,873 hectares in 2015 [13]. Over 85% of agricultural households in the country are engaged in crop production including food crops, tree crops and horticultural crops. Sierra Leone is divided into five regions and 16 agricultural districts [13]. At the regional level, the Northern Region recorded the highest proportion of households engaged in crop production (37%), followed by the Eastern Region (26%), the Southern Region (22%) and the Western Area Region (1%) [13].

The MAF is the governmental institution responsible for the regulation of pesticides. The Crop Protection Unit (CPU), which is the National Plant Protection Office (NPPO) within MAF, is divided into three main subunits: (1) Pesticide management; (2) Phytosanitary, and (3) Integrated Pest Management. The NPPO is responsible for the registration, monitoring and overall regulation of pesticides in Sierra Leone [14]. It is part of the National One Health Committee and the Antimicrobial Stewardship Technical Working Group, with a focal point representing the unit in both groups.

### 2.2. Pesticide Imports and Distribution

The pesticides imported by the MAF, and to be used in farming, are divided into several categories based on their properties: they are generally classified as insecticides (kill insects), fungicides (kill fungi) and herbicides (kill weeds). Pesticides are not only imported by the MAF, but also by the Ministry of Health & Sanitation (MoHS), Universities, non-governmental organizations (NGOs), the private sector and agricultural companies. The MAF CPU regulates the imports of agricultural pesticides. The pesticides recommended for use in Sierra Leone by the MAF are presented in Table 1.

The MAF imports pesticides based on specifications produced by CPU, including the type of pesticide, crop application, and amounts. The specifications are determined by decisions based on the review of monthly district-level reports on pest levels, on priority crops, and the size of cultivated land. MAF imports of pesticides are documented in the Pesticide Import Inventory kept by the CPU.

The MAF distributes pesticides for sale on a cost recovery basis. However, in the event of natural disasters, MAF provides pesticides for farmers free of charge. The pesticides are distributed by the CPU at the national level to the sub-offices in charge of pest management in the 16 agricultural districts. The CPU determines the distribution list based on crops’ comparative advantage and the presence of pests and diseases from monthly district reports. Farmers can access pesticides stored at the district sub-offices.

### 2.3. Regulation of Pesticides and Antimicrobials

Sierra Leone follows the pesticide regulations as outlined in the Economic Community of West African States (ECOWAS) West Africa Agricultural Policy [14]. The Sierra Leone National Pesticide Policy covers the regulation and use of pesticides from production to disposal and established the National Pesticide and Pest Management Committee (NPPMC) as the responsible agent to evaluate all pesticide applications and carry out the pre- and post-registration of pesticide regulatory functions. The National One Health Committee is responsible for ensuring the regulation of antimicrobials and strengthening the coordination and collaboration of all sectors (plant, animal, human and the environment) [15].

### 2.4. Ebola Epidemic 2014–2016

The Ebola virus disease epidemic occurred from 2014 to 2016 in Sierra Leone. During these years, there was limited government procurement or importation of pesticides as all government resources were geared towards the management of the Ebola crisis. Most farming activities were suspended due to restrictions placed on movement and grouped farming. This likely had a substantial impact on the agriculture sector with an increase in pest and disease infestations. These factors contributed to reduced food production and food insecurity. There was food scarcity for some commodities, especially rice and vegetables, and this significantly impacted the livelihoods of farmers. The MAF subsequently imported farming pesticides only in 2014 [12].

### 2.5. Study Population, Variables and Data Collection

The study population included all pesticides imported into Sierra Leone by the MAF, before (2010), during (2014), and after (2021) the Ebola epidemic in Sierra Leone. Study variables included the total amount of pesticide (kg or liters) and type, year of import, district of distribution, desired crop to be applied, type of pesticide by WHO hazard class and application type. The active amount, strength and crop application type of a specific pesticide substance were sourced from product labels. The antimicrobial substance was determined by the chemical classification of a substance. Application type was determined by the chemical classification. Data was collected from pesticide importers’ record books and from MAF Pesticide Import Inventory (types and quantity).

### 2.6. Analysis and Statistics

Secondary electronic aggregated data were cleaned and then analysed using Microsoft Excel. Summary statistics were used to quantify the amounts of pesticide importation before, during and post Ebola were calculated and compared across the years. Importation quantities were compared by district, desired crops, pesticide type and hazard level. A comparison of pesticides by region over the time period of the study was not possible because of the uncertainty of the timing of distribution and use of pesticides during the study years.

## 3. Results

During the study period, the importation of pesticides by the government was only completed in three years: 2010, 2014 and 2021. In total, 26 pesticides were purchased and these are presented in Table 2.

The total weights and volumes of importation of those pesticides for each of the three years (2010, 2014 and 2021) are presented in Table 3.

Figure 1 shows the importation of pesticides into Sierra Leone for farming crops, vegetables and fruit in 2010 (before Ebola), 2014 (during Ebola) and 2021 (after Ebola). Overall, there was an increasing trend of pesticide imports for all types. The largest increase by crop was found to be among cereals: before Ebola, 6200 L of pesticides were imported compared with 42,050 L post-Ebola. Likewise, similar increases in pesticide imports were seen for both vegetables and fruits.

The largest increase in imported pesticides occurred with Neem, an insecticide, which increased from 0 L before the epidemic to 6900 L post-Ebola. The fungicides imported into Sierra Leone in the three years are shown in Figure 2. Amongst these, the largest increase over the time period was with the importation of Mancozeb, which has only been imported since 2021.

Of note, there were no imported pesticides reported to be used before Ebola for nuts, seeds, legumes, tobacco, ornamental, or pasture crops. For each of these crops, imported pesticides were reported post-Ebola.

The regions of Sierra Leone reporting the most growth in pesticide importation over time were the Northern and Southern (see Figure 3).

When comparing all the regions, none showed a decrease in the importation of pesticides before and after Ebola. Even during the Ebola epidemic, increased importation was reported across all regions. However, there were no data available demonstrating amounts distributed over the time intervals between importations.

Non-antimicrobial pesticides were found to be imported in larger quantities compared to antimicrobial pesticides before, during and after Ebola (see Table 4). The use of antimicrobial pesticides increased dramatically from 150 kg in 2010 to 910 kg in 2014 and 23,560 kg in 2021. Within the pesticide classes, the predominantly imported types were insecticides during the entire reporting period. During the Ebola period, the quantity of imported insecticides increased marginally from 2200 L to 2315 L.

Finally, when comparing the WHO hazard class of imported pesticides, extremely hazardous pesticides were only reported before Ebola (see Table 4). The majority of pesticides that were imported across the reporting period were either moderately, slightly, or unlikely to be hazardous (see Figure 4).

## 4. Discussion

This is the first study, that we are aware of, reporting the quantities and types of farming pesticides (insecticides, fungicides and herbicides) imported into Sierra Leone by the MAF from 2010–2021. Key findings from this study include: (1) pesticide importation by the MAF has increased over this time period; (2) among pesticides, insecticide importation has increased the most; (3) importation of antimicrobial pesticides has increased, particularly with fungicides; and (4) the overall hazard class risk of imported pesticides has improved. It was reassuring that despite the Ebola epidemic there appeared to be no clear decrease in the amount of pesticides imported into Sierra Leone. In contrast, we found the opposite, there appeared to be an increase in the importation of all three major classes of pesticides by the MAF despite the impact of both the Ebola and COVD-19 epidemics. The reasons driving the increased pesticide procurement remain unclear but are likely multifactorial.

Food security has become an increasing struggle for the population of Sierra Leone, as recently reported in a comprehensive review by the World Food Programme comparing available data from 2010, 2015 and 2020. Their findings show overall food insecurity increasing from 45% (2010) to 50% (2015), to 57% (2020), and this now affects over 4.7 million people across Sierra Leone [16]. A primary driver of this food insecurity has been poor crop production caused by outdated farming practices and challenges caused directly by both the Ebola epidemic and the COVID-19 pandemic [16]. With both of these outbreaks, there has been significant disruption of agricultural inputs, workforce availability and economic stability. Between 64–81% of those working in agriculture reported that COVID-19 markedly impacted their livelihoods [16].

As previously noted, a key component of improved crop production and reduced crop losses is the appropriate and safe use of pesticides. Our study findings demonstrated increases in pesticide importation supported by the government at key times, particularly in the face of both Ebola and COVID-19 outbreaks. The distribution of pesticides was found to be higher among the northern and southern regions of the country, which coincides with a higher level of crop production in these areas. Despite the increased investment in pesticide importation, food insecurity for the Sierra Leone population has grown substantially. It is possible that there were other factors leading to increased pesticide importation that we are unaware of. These could include increased acreage farmed, improved farming technology and enhanced cultivation techniques. It is also possible that there were other sources of both legal and illegal pesticide procurement occurring that were not being tracked by the MAF.

Despite the growing importation of pesticides, legal and affordable access to these crucial agrochemicals remains limited, as does the education process on how to use them safely. A study in 2016 from Sierra Leone estimated that 60% of pesticides being used by rice farmers (rice is the most widely grown crop in the country) were obtained from illegal country entry and 71% of the farmers received no training on safe usage [11]. In addition, almost 50% of the pesticides being used in Sierra Leone originated from the neighboring Republic of Guinea, with instructions for their use written in French, which is not commonly understood in Sierra Leone, and almost 50% of pesticides were sold by street vendors, often in unlabeled containers [11]. Finally, it is highly likely that significant amounts of illegal pesticides are either counterfeit or heavily diluted, leading to poor crop production and further food insecurity [1].

AMR within plant producing industries is most likely to occur with the use of fungicides [17]. Fungal infections among humans have increased over the last 20 years, particularly among those with suppressed immune systems. Concurrent with our increasing reliance on antifungal medications, we are beginning to see increasing human pathogenic fungi with AMR [18]. Fungal infections are now estimated to cause over 2 million deaths per year, similar to the mortality rates reported for tuberculosis and malaria [19]. Widespread use of fungicides in Europe, for example, compared to the relatively low use in the United States, has been associated with a significantly higher rate of resistant fungal infections in European countries [17].

We found increasing amounts of fungicides imported over the study period in Sierra Leone. Fortunately, the predominant fungicide being imported by the government is a non-azole, which has not yet been closely associated with AMR. However, given the uncertain amount of importation through other entities (MOHS, universities, NGOs, agricultural companies and the private sector) and the illegal importation market, it remains unclear exactly how much and what types of fungicide are being used within the country. The type and amount of fungicides being imported and used within Sierra Leone should be closely monitored as these agrochemicals present some of the highest risks for AMR.

Glyphosate is one of the most commonly used herbicides globally and AMR has risen concurrently since its introduction and escalation. Our study found that glyphosate made up almost 25% of the total herbicidal imports into Sierra Leone. A theoretical relationship was proposed that suggests glyphosate “may serve as one of the drivers for antibiotic resistance” [20]. A connection between exposure to glyphosate and “adaptive multiple antibiotic resistance” in *Escherichia coli* and *Salmonella enterica* to quinolones and aminoglycosides, has subsequently been demonstrated [21,22,23]. Further research has supported these findings [24,25]. 

Despite these concerns, glyphosate remains classified as slightly hazardous by WHO. There has now been a relatively clear understanding that AMR can occur when pesticides are used indiscriminately in agricultural settings, contaminating the environment and leading to changes in the “bacterial community structure” and the induction of bacterial and fungal resistance [26].

Finally, when assessing the amount of pesticides imported into Sierra Leone over the study period by the WHO hazard class, we found that in both 2014 and 2021, there were none in the extremely hazardous group. In addition, in 2021, there was only a minimal amount of highly hazardous pesticides imported by the MAF. These findings suggest that the government is successfully guiding the use of legally imported pesticides in a safer direction. It remains uncertain why all three classes of pesticides (insecticides, herbicides, fungicides) had increased amounts imported over the study period. We suggest this was a coordinated effort to improve crop production in response to increasing food insecurity. However, there were likely other factors driving this finding and further study is necessary to identify these unknown variables. 

The major strengths of this study were having access to all of the MAF’s pesticide importation records spanning 2010 through to 2021 and following the STROBE (Strengthening the Reporting of Observational Studies in Epidemiology) guidelines statement [27]. This allowed a comprehensive review of the quantity and type of imported pesticides by region and year. The main limitation was not being able to accurately approximate the amount of pesticides made available within the country through other sources, both legal and illegal. By some accounts, there is likely to be a large quantity of counterfeit pesticides of uncertain type coming into the region [1,11]. We were also unable to clarify the time taken between purchase and distribution of the imported pesticides, thus making a completely comprehensive comparison of amounts imported/distributed not possible. 

Based on the study findings and the evolving knowledge surrounding pesticide use in both developed and developing countries, we can make several recommendations that should be considered with regard to pesticide management. Most importantly, because of the increasing food insecurity within Sierra Leone and the outmoded farming practices, it is critical to continue MAF support, in collaboration with the private and NGO sectors for the importation, distribution and monitoring of pesticide use. Monitoring and control of importation need to expand to include the private sector to ensure proper use of exclusively safe pesticides and to reduce associated health risks and the emergence of AMR. This will require substantial financial subsidization, and international sponsorship will likely be needed for success. 

Likewise, alternative approaches to the use of current pesticides within Sierra Leone should be considered. Integrated pest management is the use of natural predators, non-chemical pesticides, selection of the most specific pesticides at the lowest doses, in concert with alternating crops seasonally and anti-resistant strategies. [1] However, Integrated Pest Management faces notable challenges, especially in developing countries. These include the need for significant education and support for implementation, introduction of new non-chemical products, buy-in from the farming community and combating the economic influence of the pesticide industry [1]. Integrated Pest Management has been proposed by the MAF, but they have acknowledged the multiple challenges faced with scaling this up [28].

Other approaches to limiting the use of pesticides include shifting to organic farming techniques and agroecology. [1] These novel practices would also require additional support but could be conducted collaboratively with in-country universities leading the educational component for local farmers and cooperatives. 

From another perspective, it is equally or perhaps more important, to improve farming practices within Sierra Leone overall. The key to increased food production is simply not just better use of pesticides. This could include utilising more modern cultivation practices with advanced technology for planting and harvesting. Such improvements would require significant investment, education and support, but would likely yield higher crop production, reducing food insecurity. Enhanced farming practices could also lead to lower reliance on pesticides as well.

To reduce the risk of AMR going forward, the MAF should consider removing the currently approved azole-containing fungicides from importation. In the meantime, another rapidly conducted operational research study could shed light on the amounts and types of pesticides imported (including illegally) by the private sector and farmers and measure the magnitude of the problem. There also needs to be strong linkages between the MAF, National One Health Committee and the Antimicrobial Stewardship Technical Working Group, to assist with safe import decisions with regards to AMR risks. Finally, a country-wide initiative to improve education and knowledge surrounding the safe use of pesticides and the dangers of using non-approved agrochemicals would be beneficial.

## 5. Conclusions

In conclusion, our study found that in response to rapidly growing food insecurity from multiple causes, the government of Sierra Leone has dramatically increased the amount of less hazardous pesticide importation to improve crop production. Further support is necessary to decrease the risk of worsening food shortages and the possible threat of emerging AMR. We recommend continued monitoring and surveillance, with further studies on the most appropriate responses to these multiple challenges.

## Figures and Tables

**Figure 1 ijerph-19-04792-f001:**
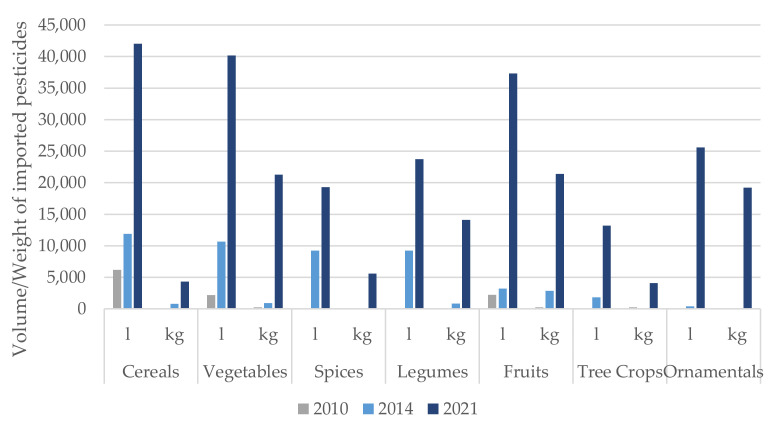
Pesticides by year and crop type applied, imported into Sierra Leone during 2010, 2014 and 2021.

**Figure 2 ijerph-19-04792-f002:**
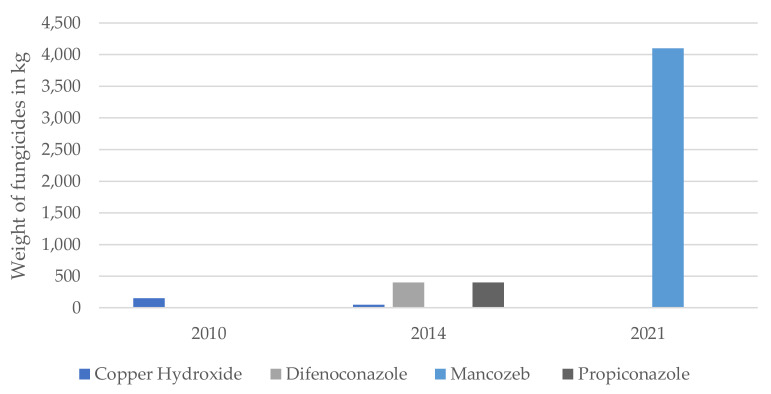
Fungicides imported for farming into Sierra Leone during 2010, 2014 and 2021 in kg.

**Figure 3 ijerph-19-04792-f003:**
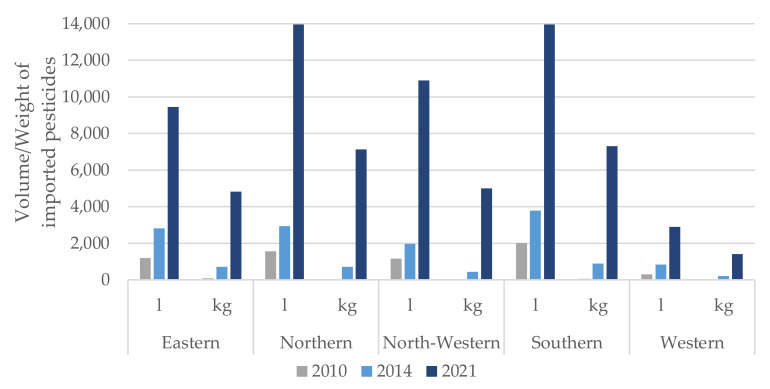
Pesticides by year and region imported for farming into Sierra Leone during 2010, 2014 and 2021.

**Figure 4 ijerph-19-04792-f004:**
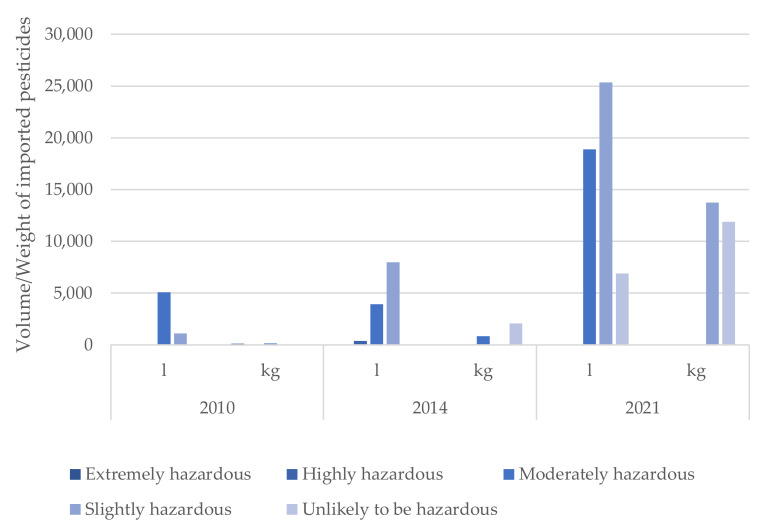
Antimicrobial pesticides (antifungal, antibacterial, antiparasitic, antiviral) per WHO hazard class imported for farming into Sierra Leone during 2010, 2014 and 2021.

**Table 1 ijerph-19-04792-t001:** Pesticides recommended by MAF for agricultural use in Sierra Leone.

Type	Active Ingredient
Insecticides	Imidacloprid
	Alpha-Cypermethrin
	Cypermethrin
	Deltamethrin
	Chlorpyrifos
	Diazinon
Fungicides	Captan
	Mancozeb
	Propineb
	Difenoconazole
	Propiconazole
	Tebuconazole
	Curpic oxide
Herbicides	Ethofumesate
	Glyphosate
	Metamitron

**Table 2 ijerph-19-04792-t002:** Active ingredients, hazard levels and types of pesticides imported into Sierra Leone during 2010, 2014 and 2021.

Pesticide Name	Active Ingredient	Hazard Level	Type	Antimicrobial Properties
Kocide 101	Copper Hydroxide	Moderately hazardous	Fungicide	Yes
Difenoconazole	Difenoconazole	Moderately hazardous	Fungicide	Yes
Propiconazole	Propiconazole: 1-[[2-(2,4-dichlorophenyl)-4-propyl-1,3-dioxolan-2-yl]methyl]-1H-1,2,4- triazole	Moderately hazardous	Fungicide	Yes
Mancozene	Mancozeb	Slightly hazardous	Fungicide	Yes
Spinosad	Spinosad (a mixture of spinosyn A and spinosyn D)	Slightly hazardous	Fungicide	Yes
Aluminum phosphide	Aluminum Phosphide	Highly hazardous	Fumigants	No
Paraquat	Paraquat dichloride (1,1′-dimethyl-4,4′-bipyridinium dichloride)	Moderately hazardous	Herbicide	No
Propanil	Propanil: 3′,4′; dichloropropionanilide	Moderately hazardous	Herbicide	No
Round up	Glyphosate, *N*-(phosphonomethyl)glycine	Slightly hazardous	Herbicide	No
Atrazine	Atrazine (2-chloro-4-ethylamino-6-isopropylamino-s-triazine)	Slightly hazardous	Herbicide	No
Rice Force	Rice force oxidiazium	Slightly hazardous	Herbicide	No
Butachlor	Butachlor	Slightly hazardous	Herbicide	No
Carbofuran	Carbofuran	Extremely hazardous	Insecticide	No
Dichlorvos	Dichlorvos	Highly hazardous	Insecticide	No
Methomyl	Methomyl (S-methyi-*N*-[(methyl carbamoyl), oxy (thioacetamide)	Moderately hazardous	Insecticide	No
Carbaryl	Carbaryl (1-naphthyl *N*-methylcarbamate)	Moderately hazardous	Insecticide	No
Manerzane	Chlorpyrifos + Abamectin	Moderately hazardous	Insecticide	No
Diazinon	*O*,*O*-diethyl-*O*-(2-isopropyl-6-methyl-4-pyrimidinyl) phosphorothioate	Moderately hazardous	Insecticide	No
Chlorpyrifos	*O*,*O*-diethyl-*O*-(2-isopropyl-6-methyl-4-pyrimidinyl) phosphonothioate	Slightly hazardous	Insecticide	No
Thuricide	Bacillus thuringiensis	Slightly hazardous	Insecticide	Yes
Lufenuru	Lufenuron	Slightly hazardous	Insecticide	No
Eradicoat	Maltodextrin	Slightly hazardous	Insecticide	No
Neem	Azadirachtin	Unlikely to be hazardous	Insecticide	No
B. Bassiana	Beauveria bassiana Strain GH	Unlikely to be hazardous	Insecticide	Yes
Metarhizum anisopliae	Metarhizium anisopliae strain F52	Unlikely to be hazardous	Insecticide	Yes
Methyl eugenol	Benzene, 1,2-dimethoxy-4-(2-propenyl)	Unlikely to be hazardous	Pheromone	No

Source: The WHO Recommended Classification of Pesticides by Hazard and Guidelines to Classification, 2019 ed.

**Table 3 ijerph-19-04792-t003:** Quantities of pesticides imported for farming into Sierra Leone during 2010, 2014 and 2021.

Pesticide	Pre Ebola (2010)		During Ebola (2014)		Post-Ebola (2021)	
	Quantity	Unit	Quantity	Unit	Quantity	Unit
Kocide 101	150	kg	50	kg	-	kg
Difenoconazole	-	kg	400	kg	-	kg
Propiconazole	-	kg	400	kg	-	kg
Mancozene	-	kg	-	kg	4100	kg
Spinosad	-	kg	-	kg	4350	kg
Aluminum phosphide	-	kg	50	kg	54	kg
Paraquat	-	L	1845	L	-	L
Propanil	4000	L	668	L	5400	L
Round up	-	L	500	L	4100	L
Atrazine	-	L	6000	L	-	L
Rice Force	-	L	1000	L	5600	L
Butachlor	-	L	-	L	4450	L
Carbofuran	100	kg	-	kg	-	kg
Dichlorvos	-	L	400	L	-	L
Methomyl	-	l	915	L	-	L
Carbaryl	-	L	-	L	4550	L
Manerzane	-	L	-	L	4550	L
Diazinon	1100	L	500	L	4400	L
Chlorpyrifos	1100	L	500	L	2900	L
Thuricide	-	kg	-	kg	5300	kg
Lufenuru	-	L	-	L	2800	L
Eradicoat	-	L	-	L	5500	L
Neem	-	L	-	L	6900	L
B. Bassiana	-	kg	-	kg	5600	kg
Metarhizum anisopliae	-	kg	60	kg	4210	kg
Methyl eugenol	30	L	-	L	-	L
	-	kg	2000	kg	2060	kg
**Total**	6230	L	12328	L	51150	L
	250	kg	2960	kg	25674	kg

kg = kilogram, L = liters.

**Table 4 ijerph-19-04792-t004:** Quantities of antimicrobial pesticides by application type and hazard class imported for farming into Sierra Leone during 2010, 2014 and 2021.

Total		Pre Ebola (2010)		During Ebola (2014)		Post-Ebola (2021)	
		Quantity	Unit	Quantity	Unit	Quantity	Unit
Type of pesticide	Antimicrobial	150	kg	910	kg	23,560	kg
	Non-antimicrobial	6230	L	12,328	L	51,150	L
		100	kg	2050	kg	2114	kg
Class of pesticide	Fungicide	150	kg	910	kg	4100	kg
	Insecticide	2200	L	2315	L	31,600	L
		100	kg		kg	21,520	kg
	Herbicide	4000	L	10,013	L	19,550	L
	Fumigant	-	kg	50	kg	54	kg
	Pheromone	30	L	-	L	-	L
		-	kg	2000	kg	-	kg
WHO Hazard class	Extremely hazardous	100	kg	-	kg	-	kg
	Highly hazardous	-	L	400	L	-	L
		-	kg	50	kg	54	kg
	Moderately hazardous	5100	L	3928	L	18,900	L
		150	kg	850	kg	-	kg
	Slightly hazardous	1100	L	8000	L	25,350	L
		-	kg	-	kg	13,750	kg
	Unlikely to be hazardous	30	L	-	L	6900	L
		-	kg	2000	kg	11,870	kg

kg = kilogram, L = liters.

## Data Availability

The dataset used in this paper has been deposited at figshare.com and is available under a CC BY 4.0 license.

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
