# Peer review of "Pesticide Importation in Sierra Leone, 2010–2021: Implications for Food Production and Antimicrobial Resistance"

_ijerph, 2022, doi:10.3390/ijerph19084792_

Round 1

Reviewer 1 Report

These are my main comments on the manuscript (IJERPH-1626973) entitled “Pesticide importation in Sierra Leone, 2010-2021: implications 2 for food production and antimicrobial resistance”. The manuscript examines the imported pesticides in Sierra Leone between 2010 to 2021. Following moderated revisions should be incorporated in the manuscript prior to acceptance.
L.36: Delete “In summary”
Ls.42-43: Keywords should be in alphabetic order. Also, keywords serve to widen the opportunity to be retrieved from a database. To put words that already are into title and abstracts makes KW not useful. Please choose terms that are neither in the title nor in abstract.
L.51: Which pesticides? Antibiotics? Antifungal products? Explain
L.66: change “recognises” to “recognizes”
L.76: change “utilising” to “utilizing”
Ls.98-99: Hypothesis should be before the main objective. Place in line 94.
Table 1 and results: How do herbicides and insecticides cause antimicrobial effects? Is this information important in your results?
Ls.164-168: The statistical analysis used in this study is unclear. Explain.
Figures 1, 2, 3, and 4: Are the data in the columns means? Accumulated data? Place information in y-axis.

Author Response

Dear Editors:

We greatly appreciate the time spent reviewing our manuscript by the reviewers and their valuable insights and comments. Please see our point-by-point response the reviewers’ comments as requested. These can also be found within the track changes of our manuscript and in the accompanying “clean” version. Please let us know if there were any omissions.

Reviewer 2 Report

Comments to the Authors

General comment:

I do not understand the comparative advantage of work? All data show an annual increase in cache consumption. This can be caused by many things and is not just related to the Ebola epidemic. All possible aspects must be stated. For example, in all these 11 years, it is possible that due to the advancement of technology and increasing productivity and cultivation on the scale of each meter of agricultural land, the volume of cultivation has increased, and consequently the consumption of pesticides has increased. As an important part of the study, I would like the authors to introduce the reasons for the increase in consumption of all three categories of pesticides, and finally to summarize their results by introducing solutions and alternative control methods.

Finally any related Recommendations? If we want to reduce the consumption of these substances, especially those that have antimicrobial properties, what alternatives will we have? In my opinion, at least two paragraphs in the discussion of the article should talk about these cases and, considering the situation in their country, authors should interpret and propose the existing solutions.

 Keywords: please delete “sort it”. Mesh terms should be used as keywords.

Introduction

Related to this part (line 58-60): “To reduce pesticide risks, countries must wisely manage their use of pesticides…. Please describe the characteristic pesticide risks in the introduction (five types of pesticides).

Is there any local producer/company in the country? If yes, it is better to use their data, too.

Related to this part (line 74-76): “Pesticide usage in sub-Saharan African countries is thought to be limited, with a reported average of 16% …. I would like to read some related data about the rate of using the three types of pesticides (insecticides, fungicides, and herbicides) that were used in the country.

Method

Much more detail should be provided in the “Variables and Data Collection” section.

Result

Related to this part: The regions of Sierra Leone reporting the most growth in pesticide importation over 200 times were the Northern and Southern…..   To make the data more useful for regional readers, I suggest that the reasons for consuming more pesticides in these sections (the Northern and Southern) be examined and introduced.

At first, according to the author, it was thought that the purpose of the study was to determine the use of pesticides and the impact of epidemics on it, but according to the results, the incidence of Ebola had little effect on it and pesticide use has increased continuously annually. How is it interpreted?

Best

Author Response

(The authors gave the same response as above.)
